# PCSK9 as a Target for Development of a New Generation of Hypolipidemic Drugs

**DOI:** 10.3390/molecules27020434

**Published:** 2022-01-10

**Authors:** Nikolay Kuzmich, Elena Andresyuk, Yuri Porozov, Vadim Tarasov, Mikhail Samsonov, Nina Preferanskaya, Valery Veselov, Renad Alyautdin

**Affiliations:** 1Laboratory of Drug Safety, Smorodintsev Research Institute of Influenza, WHO National Influenza Centre of Russia, 15/17 Professor Popov Street, 197376 Saint-Petersburg, Russia; nikolay.kuzmich@influenza.spb.ru; 2World-Class Research Center “Digital Biodesign and Personalized Healthcare”, I.M. Sechenov First Moscow State Medical University, 8/2 Trubetskaya Street, 119991 Moscow, Russia; porozov_yu_b@staff.sechenov.ru; 3Department of Computational Biology, Sirius University of Science and Technology, Olympic Ave. 1, 354340 Sochi, Russia; 4Department of Pharmacology, I.M. Sechenov First Moscow State Medical University, 8/2 Trubetskaya Street, 119991 Moscow, Russia; tarasov-v-v@mail.ru (V.T.); mikesamsonov@yahoo.com (M.S.); Ninusik50@yandex.ru (N.P.); sacrednibelung@yandex.ru (V.V.); alyautdin_r_n_1@staff.sehenov.ru (R.A.); 5Scientific Centre for Expertise of Medicinal Application Products, 8 Petrovski Boulevard, 127051 Moscow, Russia

**Keywords:** PCSK9, LDLR, antihyperlipidemic therapy, lipid-lowering drugs, PCSK9 inhibitors

## Abstract

PCSK9 has now become an important target to create new classes of lipid-lowering drugs. The prevention of its interaction with LDL receptors allows an increase in the number of these receptors on the surface of the cell membrane of hepatocytes, which leads to an increase in the uptake of cholesterol-rich atherogenic LDL from the bloodstream. The PCSK9 antagonists described in this review belong to different classes of compounds, may have a low molecular weight or belong to macromolecular structures, and also demonstrate different mechanisms of action. The mechanisms of action include preventing the effective binding of PCSK9 to LDLR, stimulating the degradation of PCSK9, and even blocking its transcription or transport to the plasma membrane/cell surface. Although several types of antihyperlipidemic drugs have been introduced on the market and are actively used in clinical practice, they are not without disadvantages, such as well-known side effects (statins) or high costs (monoclonal antibodies). Thus, there is still a need for effective cholesterol-lowering drugs with minimal side effects, preferably orally bioavailable. Low-molecular-weight PCSK9 inhibitors could be a worthy alternative for this purpose.

## 1. Introduction

Atherosclerosis is the leading cause of numerous cardiovascular problems such as coronary artery disease, stroke, hypertension, arrhythmia, angina pectoris, and heart valve problems. Changes in lipoprotein and blood lipid levels generally show a consistent relationship with the risk of myocardial infarction. A higher atherogenic lipid load (as measured by the concentration of total cholesterol, LDL, or apolipoprotein B; or the ratio of total cholesterol to high-density lipoprotein or apolipoprotein B toapolipoprotein A1) is associated with a gradual increase in the incidence of cardiovascular disease. In addition, lowering the LDL cholesterol has been found to reduce the risk of cardiovascular disease, which in turn is consistent with the range of pretreatment LDL cholesterol levels [1,2,3,4,5,6]. The rate of reduction in cardiovascular disease is proportional to the rate of decrease in LDL cholesterol, and it is estimated that a 1 mmol decrease in LDL cholesterol in middle-aged adults over 5 years leads to a 20% reduction in the risk of cardiovascular disease [2]. The continuously unmet medication needs remain the main driver for the development of new drugs in this area. Research is currently focused on drug discovery based on new targets.

Proprotein convertase subtilisin/kexin type 9 (PCSK9) inhibitors are the substances which are aimed at protecting liver low-density lipoprotein receptors (LDLRs) from degradation.

PCSK9 is a hydrolase/serine protease produced by a gene of the same name. The PCSK9 gene consists of 12 exons and 11 introns. Its expression is initiated by sterol regulatory element-binding protein 2 (SREBP-2) transcription factor and another activator, hepatocyte nuclear factor 1 alpha (HNF1A), which is liver tissue-specific. The transcription followed by splicing leads to the formation of PCSK9-mRNA.

Translation leads to the formation of immature predecessor pre-pro-PCSK9 consisting of four domains: signal peptide, prodomain, N-terminal domain, catalytic, and cysteine/histidine-rich domain (C-terminal domain) [7].

After post-translational modifications including folding, partial hydrolysis, and glycosylation occurring in the endoplasmic reticulum (ER) and Golgi apparatus, the mature PCSK9 protein is formed.

An understanding of the PCSK9 formation stages is of paramount importance for studying the possible therapy targets in the future. Indeed, to date, some drugs have been developed that are capable of inhibiting PCSK9 biosynthesis during its transcription as well as post-transcriptional modifications such as antisense nucleotides and small interfering RNAs (siRNA) [8].

The transformation of pre-pro-PCSK9 into pro-PCSK9 is carried out in the ER by eliminating the 30 amino acids long signal peptide. Next, the prodomain is separated from the pro-PCSK9 autocatalytically and, in turn, binds to the catalytic domain, forming the PCSK9–prodomain complex. This prodomain assists in blocking PCSK9 enzymatic activity and provides the correct folding necessary for PCSK9 transportation from the ER to the Golgi apparatus [9,10,11]. Strom and his colleagues established [9] that pro-PCSK9 is able to interact with LDLR in the ER, located on the ribosomes during their post-translational modifications.

Here we observe the following picture: r-LDL promotes autocatalytic prodomain removal from pro-PCSK9, then the autocatalytically cleaved PCSK9 acts as a chaperone providing the correct folding of ribosomal LDL [9]. During the transportation from the ER to the Golgi apparatus, the PCKS9–prodomain complex attached to the catalytic domain blocks the binding of other proteins to the prodomain, inhibits its enzymatic activity, and protects it from interactions with enzymes, especially with furine. Two mutations (Ala443Thr and Cys679X) decrease PCSK9 stability towards furin, which in turn leads to a shorter life span of the PCSK9. This is accompanied by a decrease in the LDL level in the serum and, consequently, a decreased risk of atherosclerosis. Therefore, it can be deduced that furin modulators could be useful for hyperlipidemic therapy [10,12,13,14,15].

The SEC24A protein promoting the formation of transporting vesicles is also an important element in the transportation of PCSK9 to the Golgi apparatus. Thus, development of SEC24A inhibitors [11] can also be useful in PCSK9 modulation.

Glycosylation and the ultimate elimination of PCSK9–prodomain occur in the Golgi apparatus, resulting in the formation of mature PCSK9 [16,17,18,19,20,21]. In the Golgi apparatus, PCSK9 covalently binds to sortilin. Gustafsen and her coauthors [22,23,24,25,26] suppose that sortilin is vital for PCSK9 transportation to the membrane followed by its secretion into the blood. The high correlation discovered between serum sortilin and PCSK9 levels enables the suggestion that the PCSK9 level is decreased by sortilin inhibition [27,28].

The LDLR itself consists of several independent domains and motifs, including the ligand-binding domain, the epidermal growth factor precursor (EGFP) homology domain, the highly glycosylated domain, the transmembrane domain, and the cytosolic C-tail. The first three domains are the extracellular ones. 

The LDLR is synthesized in almost all the cells of an organism on the ER ribosomes, and then it is modified in the Golgi apparatus before being transported to the membrane within vesicles. The activation and inhibition of its transcription is regulated by the cholesterol level in a cell. Its excess blocks the transcription, and vice versa.

The membrane-bound LDLR is localized at clathrin-coated pits. The two basic endogenous LDLR ligands are apolipoprotein B (apoB-100) and apolipoprotein E (apoE). Upon binding to endogenous ligand, the LDLR binds then to LDL cholesterol via adaptin, and the LDLR-containing caveola forms a vesicle and leaves the cell surface, travelling inwards by endocytosis. After that, the LDLR is either degraded in lysosomes or returns back to the membrane. About 70% of LDLs are absorbed via receptor-mediated endocytosis by liver cells but excessive amount of LDL significantly increases risk of atherosclerosis because of their non-specific accumulation on the blood vessel walls. The LDLR is one of the major regulating elements of serum lipid metabolism.

## 2. Physiological Role of PCSK9

One of the most important PCSK9 functions known today is without a doubt its participation in LDLR expression regulation, leading to regulation of the LDL cholesterol level in the blood. Additionally, PCSK9 carries out two quasi-independent functions: on the one hand, it decreases the LDLR density on hepatocytes’ surface, and on the other hand, it prevents the reverse capture of the newly synthesized VLDL, letting them reach the peripheral tissues. However, it should be kept in mind that PCSK9 is able to interact with other receptors and thus may possess a far broader spectrum of activity which has not been completely investigated to date.

## 3. Mechanism of Regulation of PCSK9-Mediated LDLR Expression in Hepatocytes

The clearance and catabolism of serum LDL occurs in the liver. These are the hepatocytes which are major regulators of the LDL level, expressing LDLRs which capture the LDLs and remove them from the blood plasma. The LDL/LDLR complex enters a hepatocyte within clathrin vesicles, gradually merging with the endosomes. The endosomal acidic medium activates a complex dissociation and the released LDLRs return to the hepatocyte surface, continuing their job of removing LDL from the serum [29]. It is known that the calcium-mediated interaction of the PCSK9 catalytic domain with the A-EGFP repeat of LDL takes place on the hepatocytes’ surface. After the LDL binds to the LDLR, the LDL/LDLR/PCSK9 complex relocates into the cell within the clathrin vesicle. The low-pH medium in the endosomes promotes the separation of the LDL from the complex and increases the PCSK9/LDLR binding due to additional ion–ionic interactions between the PCSK9 prodomain and the LDLR. As a result, the PCSK9 holds the LDLR in the open conformation like a pillar, without acting proteolytically, and prevents the transition in the closed conformation that is necessary for the return to the cell surface [30,31,32], which leads to a decrease in receptor expression on the hepatocyte surface and, accordingly, a decrease in LDL clearance from the plasma. Consequently, PCSK9 plays a very important role in the serum LDL cholesterol level regulation.

## 4. PCSK9-Dependent LDLR Degradation

The mechanism of PCSK9-enhanced LDLR degradation is still not quite clear. The structure of the PCSK9/EGF-A complex is consistent with the fact that affinity of PCSK9 to LDLRs increases at low pH level, which prevents the receptor from recycling. Nevertheless, the binding of PCSK9 to LDLRs can change the LDLR conformation promoted by acidic medium. Thus, the binding of PCSK9 near the EGF-A N-terminal area may affect the known interdomain interactions, either via EGF-A or via the steric effects of PCSK9 itself, or influence the EGF-A conformation and the calcium ion coordination. The intramolecular interactions in EGF-A include the interdomain packing of EGF-A with EGF-B, which is essential for LDLR stability [33,34,35], as well as the interaction of EGF-A with the ligand-binding module R7, responsible for the stiff conformation of this LDL part in a broad pH range [36,37,38,39,40,41,42]. This rigidity seems to favor the pH-dependent closed conformation which enables ligand release and LDLR recuperation from the endosomal compartment [43]. It should be kept in mind that PCSK9 can bind to an unknown factor promoting LDLR degradation.

Thus, the determination of the PCSK9/EGF-A complex structure creates a theoretical basis for the development of new alternative approaches to inhibiting PCSK9-mediated LDLR degradation (Table 1). Its analysis enables us to determine the major strategies of PCSK9 inhibition. There are several approaches known today. According to Tsuyoshi Nozue [44], the main directions are:

(1) Blocking of PCSK9/LDLR binding using mAb, adnectines, or either PCSK9/LDLR binding site mimetic peptides or small molecules.

(2) Inhibiting PCSK9 synthesis and expression in vivo using CRISPR/Cas9-based genome-editing technology, antisense oligonucleotides (ASO), or siRNA.

(3) Interference with PCSK9 secretion from the ER.

(4) Increase in PCSK9 plasma clearance.

The mAbs are efficient inhibitors of PCSK9. An elevated LDL concentration in the blood results mainly from disbalance between the synthesis and catabolism of cholesterol and disrupts its clearance. PCSK9 inhibitors should prevent cholesterol from residing on the cellular surface.

The mechanisms of action of the PCSK9 inhibitors are illustrated in Figure 1.

LDL cholesterol targets:

(1) The cholesterol-lowering agents are demonstrated on the beige hepatocyte. Both bempedoic acid (1) and statins (2) inhibit cholesterol synthesis in hepatocytes, decreasing the free cholesterol level;

(2) Ezetimib (3) inhibits the Niemann–Pick C1-like 1 protein (NPC1L1), decreasing the intestinal absorption of cholesterol from food and gall, which significantly cuts down on the delivery of cholesterol chylomicrons to the hepatocytes via portal circulation; PCSK9-specific mAbs (4) bind to PCSK9 in the blood flow. Inclisiran (5) targets PCSK9-mRNA and prevents its expression.

Both approaches decrease the circulating PCSK9 level, diminishing LDLR lysosomal degradation. All the strategies eventually enhance hepatocytic LDLR expression. Ezetimib and its active glucuronide are localized on the brush border of enterocytic microvilli, where they inhibit the steroid transporter NPC1L1, which assists intestinal cholesterol absorption [46]. The inhibition of NPC1L1 limits cholesterol absorption, thus decreasing the cholesterol stock in the liver and automatically increasing LDLR expression (as do statins and PCSK9 inhibitors). So, liver LDL absorption decreases and plasma LDL cholesterol levels sink.

As mentioned above, the PCSK9-specific mAbs can be used in therapy [47]. Several anti-PCSK9 antibodies have been tested in clinical trials. Evolocumab and alirocumab are absolutely humanized antibodies, whereas bococizumab is partially humanized. By a major part of the clinical study participants the neutralizing antibodies were produced. This circumstance led to the premature termination of bococizumab’s clinical trial [48]. In more detail, alirocumab and evolocumab have been reported to be effective in the randomized, double-blind, placebo-controlled phase III trials (ODYSSEY OUTCOMES and FOURIER (Further cardiovascular Outcomes Research with PCSK9 Inhibition in subjects with Elevated Risk), respectively) [49,50,51]. The efficacies of alirocumab and evolocumab were tested on 18,924 and 27,564 patients, respectively, who were diagnosed with atherosclerotic cardiovascular disease and received statin therapy. The administration of both antibodies resulted in a decreased LDL cholesterol level as well as a reduced rate of cardiovascular events such as myocardial infarction or stroke. Although no statistically significant decreases in the mortality rate were observed, longer follow-up terms could change the overall picture. Both drugs were well tolerated.

Keeping in mind the significant expensiveness of mAb-based therapeutics, we will focus on the economically more feasible methods of inhibition in this review. So, for example, it is known that the degradation of PCSK9 can be furin- and protein-convertases-meditated. After furin-mediated cleavage, PCSK9 is able to bind to the LDLR but its activity is decreased two-fold. Mice injected with furin-treated PCSK9 had an increased LDL level and a decreased LDLR expression.

Other proteins also affect the PCSK9 level in the serum. For example, annexin A2 localized in various cell compartments can bind by its N-terminal repeat R1 to the C-terminal PCSK9 domain, so this annexin can also be considered as endogenous PCSK9 inhibitor.

The strong correlation between serum sortilin and PCSK9 levels allows the suggestion of sortilin inhibition in order to decrease PCSK9 levels [46].

One of the rational and safe approaches to decreasing LDL cholesterol is the inhibition of PCSK9 transcription and post-transcriptional modifications, namely, using antisense oligonucleotides and siRNAs, such as inclisiran [52]. The inclisiran therapy is one of the most advanced methods of suppressing PCSK9 expression [53]. This synthetic siRNA conjugated with the branched N-acetylgalactosamine derivative L96 decreases PCSK9 expression. It targets intracellular PCSK9 synthesis and, unlike systemic mAb, binds to circulating extracellular PCSK9 [54]. The mechanisms of PCSK9 inhibition by mAbs and siRNA are illustrated in Figure 2.

Low-molecular PCSK9-targeting therapeutics are in high demand, especially taking into account their low cost and the simplicity of administration. Several projects for developing low-molecular PCSK9 inhibitors have been reported. Pfizer has reported a low-molecular inhibitor of PCSK9 secretion [56].

Nilotinib has drawn special attention. 4-S-Methyl-N-[3-(4-methyl-1H-imidazol-1-yl)-5-(trifluoromethyl)phenyl]-3-[[4-(3-pyridinyl)-2-pyrimidinyl]amino]-benzamide was chosen as a Src tyrosine kinase and protein kinase B inhibitor, available as an antitumor drug. PCSK9 inhibition by any of these compounds can lead to severe side effects [57]. Thus, the replacement of the nilotinib pyridine ring by phenyl group decreased the kinase-inhibiting activity, and in some cases only micromolar c-Kit inhibition was achieved. Meanwhile, the strongest protein–protein interaction antagonists are known to have substituents in the right benzene ring situated in metaposition to the imidazolic ring. Acidic or nonbasic hydrophilic substituents were found to be comparable with nilotinib in terms of efficacy, albeit they decreased the affinity to Abl, PDGFR-β, and Src kinases. Abl-inhibiting activity increased when trifluoromethyl group was replaced by aminosulfonic acid residue. Another nilotinib analogue was developed where the pyridine ring was replaced by the benzene ring and the trifluoromethyl was exchanged for the acetyl group. This compound has demonstrated good inhibition comparable to emalocumab.

The substitution of the aminopyrimidine linker with aryl esters demonstrated a significant increase in the inhibiting activity.

The isoquinoline ester stands apart, having yielded the strongest PCSK9/LDLR inhibition by a small molecule to date, albeit in a narrow dynamic range. Studies have shown that in combination with biaryl ester moieties, the phenoxyphenyl ester provides a perfect balance between effectivity and selectivity. The additional effectiveness of the isoquinoline group is absolutely consistent with the possible interaction between the sp2-ring nitrogen and the lysine side chain.

It should be noted that compounds from these series regenerate the LDL capture even for PCSK9 D374Y mutant which binds to LDLR with 6–25-times the strength of the wild-type PCSK9 analogue [37]. Perhaps the inner-ring angular methyl group provides a favorable change in the aromatic group orientation, resulting in a donor–acceptor hydrogen bond shorter than 3 Å.

Phenoxyphenyl esters (POPEs) bind directly to PCSK9, providing an ideal balance between affinity and selectivity. It should also be noted that these compounds, especially the amino-containing POPEs, recover the LDL capture of HepG2 cells. The PCSK9 binding with ligands in vitro is consistent with the protein–ligand interaction within the series. However, many factors of complex cellular systems must be taken into account when trying to translate the assay results. An amino group bearing POPEs restored LDLR expression on the human hepatocytes’ surface also for PCSK9 D374Y mutant. A comparative binding study including the humanized mAb alirocumab also confirmed PCSK9-meditated LDLR protection from degradation, restoring their expression on the cellular surface [37].

This study also demonstrated a decrease in the overall cholesterol level in the serum of wild-type mice. Wild-type mice are often considered to be a suboptimal choice for the modeling of LDL cholesterol-decreasing therapy because, unlike humans, the major part of their overall cholesterol is comprised of HDL fractions.

The results demonstrated the possibility for a low-molecular inhibitor to be accumulated in the serum on a sufficient level without it binding to plasma proteins other than PCSK9, thus decreasing the overall cholesterol level. These statements of course need additional studies for confirmation.

Nevertheless, it can already be claimed that, according to the array of accumulated results, the inhibition of PCSK9 can lead to a significant drop in the serum cholesterol level in vivo [37]. Thus, the concept of PCSK9 inhibition by small molecules is viable and promising. It is still unclear just how to involve the catalytic domain when it first forms a complex with its prodomain, but it becomes clear that the study of compounds involving PCSK9 for the disruption of its binding to LDLRs can have a real therapeutic application.

Min et al. [58] chose 100 compounds as possible PCSK9 binders via the in silico screening of the ChemBridge database using GOLD docking. The activity was measured by PCSK9–LDLR binding analysis, immunoblotting, and LDL cholesterol absorption analysis in vitro, as well as lipoprotein serum LC measurement in vivo.

It was demonstrated that certain small molecules dose-dependently decreased PCSK9 binding with EGF-AB LDLRs and also significantly increased the LDL level, which was confirmed by the increased absorption of fluorescent-labelled LDLs by the HepG2 cell. Additionally, one compound significantly decreased the overall cholesterol and the LDL cholesterol in the serum of wild-type mice, although this effect had not been observed before.

The PCSK9 site comprised by amino acid residues 367–381 which is situated on the PCSK9/LDLR binding interface, was considered to be targeted by inhibitors. After the removal of the EGFA domain, the PCSK9 structure was prepared by the removal of water and addition of hydrogen. The docking scores of the ChemBridge Express substances (~450,000) were computed using GOLD 4.0.1 software. One hundred of the best compounds with the highest docking scores from the ChemBridge Express collection were selected using GOLD docking. To validate the virtual screening as an instrument for searching for new PCSK9 inhibitors, the following tests were performed: in vitro experiments on HepG2 cells and binding assays using the PCSK9–LDLR binding kit. LDL cholesterol absorption analysis was performed using fluorescent-labelled Dil-LDLs and immunoblotting was performed for PCSK9 and LDLRs in HepG2 cells. The structures of the PCSK9 inhibitors are depicted in Figure 3.

CB-36 had the highest docking score. The concentrations of the tested compounds were found experimentally and were set to be as low as possible. All three compounds dose-dependently increased the LDLR and PCSK9 levels.

It was demonstrated that CB-36 and its analogues increase the LDL cholesterol capture by HepG2 cells, despite the simultaneous increase in the PCSK9 level and the sinking of the LDL fraction level.

This study demonstrated the success of the in silico approach for searching for PCSK9/LDLR blockers.

W. Petrilli, and their colleagues [59] discovered a ligand binding to an unprecedented allosteric pocket between the catalytic and C-terminal domains.

The optimization of these two initial hits using two different strategies led to high-affinity PCSK9 binders. Direct binding to the target was demonstrated in cell lysate using thermal shift.

The analysis of the PCSK9/LDLR binding interface showed the difficulty of finding low-molecular ligands for PCSK9, since both PCSK9 and the LDL-binding EGFA domain form fairly flat hydrophobic surfaces (Figure 4A). Additionally, both proteins form antiparallel beta-sheets (Figure 4A), and this circumstance made the binding of small molecules more difficult. The residues essential for the PCSK9 and LDLR interactions are marked with asterisks in Figure 4.

Another strategy was aimed at identifying small molecules able to bind to PCSK9 regardless of the site or pharmacological effect. To achieve this, mass-spectrometry-based technology (AS/MS) was employed to screen a library of over 200,000 compounds for PCSK9. According to the typical AS/MS procedure, the mixture of compounds was pre-incubated with PCSK9 and passed through the column for size-based separation. The ligands bound to PCSK9 were consequently separated by reverse-phase chromatography, following mass-spectrometric detection and identification.

The screening yielded 86 compounds which selectively bind PCSK9 via beta-lactoglobulin, which was employed as a control protein to identify non-specific binders [60,61]. Later studies showed that only compound one achieved stable and reproducible binding, as revealed by analyses including surface plasmon resonance assay [59].

Compound one showed a melting point shift 2 °C in differential scanning calorimetry (DSC), which measures the change in protein thermostability upon ligand binding (Figure 4C). Thus, this compound favored PCSK9 thermostability.

For a better understanding of the interaction details between ligand one and PCSK9, the X-ray structure of the complex was obtained. It is worth noting that this ligand binds to a previously unknown pocket located between the catalytic and C-terminal domains (Figure 4D). Interestingly, it was one of the PCSK9 sites which could be liganded in our estimation (Figure 4B). Furthermore, this druggable allosteric site is situated next to several gain- and loss-of-function mutation sites. 

The X-ray structure of PCSK9 complex with compound one revealed several potentially crucial interactions (Figure 1E). Actually, the isoquinoline amine of compound one formed hydrogen bond with the D360 residue (d = 2.4 Å) of the catalytic domain. Furthermore, the two C6 and C7 aryl methoxy groups formed hydrogen bonds with the Arg357 (d = 2.8 Å) of the catalytic domain. The Sp2 oxygen of the C1 amide, on the contrary, interacted with the Arg 458 (2.4 Å) of the PCSK9 C-terminal domain. The two enantiomers were separated by the standard methods of chiral separation, yielding compounds two and three.

The R-enantiomer (2) had a higher melting point shift (2.0 °C) compared to that of the S-enantiomer (3) (0.1 °C), so it was chosen for further SAR studies.

Initially, the efforts were focused on the stabilization effect of compound **2** on the PCSK9 C-terminal domain. So, the SAR was studied in more detail on the C1 amide; however, none of the modifications of this structure fragment led to a gain in activity. Besides, the interaction of the isoquinoline ammonium group with the D360 residue of the catalytic domain was critical for the PCSK9 binding, because the attempts to replace or remove this amino group were unfruitful. In aggregate, these results revealed that the interactions with the C-terminal and catalytic domains play a key role in the observed stabilization effect of compound two, and they should be preserved in future SAR studies.

Referring to the above, the major efforts were focused on the SARs of the C6 and C7 substituents. Multiple structures were synthesized, carrying various aromatic substituents at position C7 of the isoquinoline cycle, followed by a DSC analysis. Adding the polar functional group to the esters at C7 (carboxylic acid 4) led to a melting point shift of 3.9 °C compared with compound two. The X-ray structure of this complex revealed a salt bridge between the carboxylate and the R476 side chain guanidinium group (Figure 4F). The attempts to enhance this interaction led to the synthesis of metafluoro carbonic acid 5, which resulted in an additional PCSK9 melting point shift of 1.2 °C. The structure of PCSK9 cocrystallized with compound five revealed an additional hydrogen bond between the fluorine atom in metaposition and the backbone NH group of R476 (Figure 4G).

Furthermore, the interactions of R357 with the substituents of the C6 cycle position were considered in more detail. Although it was found that many functional groups were tolerated at this position, compound six with the tetrahydropyranyl fragment demonstrated an increased melting point shift (+1.3 °C) compared to the fluoro-substituted analogues.

In addition to the previously identified key interactions, the X-ray structure of the PCSK9–ligand eight complex revealed a new hydrogen bond between the tetrahydropyran ring and the R357 residue (Figure 5) [59]. Furthermore, a kinetic target-guided liganding strategy was employed, which used the protein as a template for 1,3-dipolar cycloaddition between two building blocks in proximity. Alkaline media lowered the activation barrier, enabling 1,2,3-triazole formation without using metallic catalysts (Figure 5 and Figure 6). Using this protein-template-based approach for controlled reactivity, optimized inhibitors were found for several classes of targets and high selectivity was reached.

To conclude, the optimization of triple-bond-containing ligands involved new substituents in the C6 and C7 positions. With CETSA, it was found that carbonic acid 9 possessed a high affinity to PCSK9 in the Huh7 cells’ lysate.

Thus, the most promising way of searching for PCSK9 inhibitors today is searching for small molecules, since it is economically feasible and enables computer-aided screening and design.

## 5. Conclusions

PCSK9 has now become an important target to create new classes of lipid-lowering drugs. The prevention of its interaction with LDL receptors allows an increase in the number of these receptors on the membrane surface of hepatocytes, which leads to an increase in the uptake of cholesterol-rich atherogenic LDL from the bloodstream.

The PCSK9 antagonists described in this review belong to different classes of compounds, may have a low molecular weight or belong to macromolecular structures, and also demonstrate different mechanisms of action. The mechanisms of action include preventing the effective binding of PCSK9 to LDLR, stimulating the degradation of PCSK9, and even blocking its transcription or transport to the plasma membrane/cell surface.

Although several types of antihyperlipidemic drugs have been introduced to the market and are actively applied in clinical practice, they are not devoid of drawbacks such as well-known side effects (statins) or high costs (monoclonal antibodies). Thus, there is still a need for effective cholesterol-lowering drugs with minimal side-effects, preferably orally bioavailable. The low-molecular PCSK9 inhibitors would be a worthy alternative for this purpose.

## Figures and Tables

**Figure 1 molecules-27-00434-f001:**
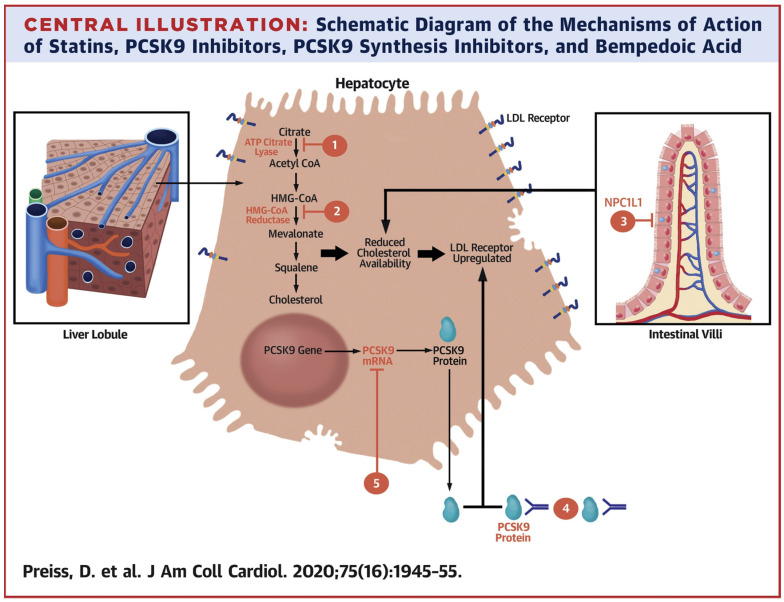
Mechanisms of action of statins, PCSK9 inhibitors, PCSK9 biosynthesis inhibitors, and bempedoic acid (Preiss, D. et al., 2020) [45].

**Figure 2 molecules-27-00434-f002:**
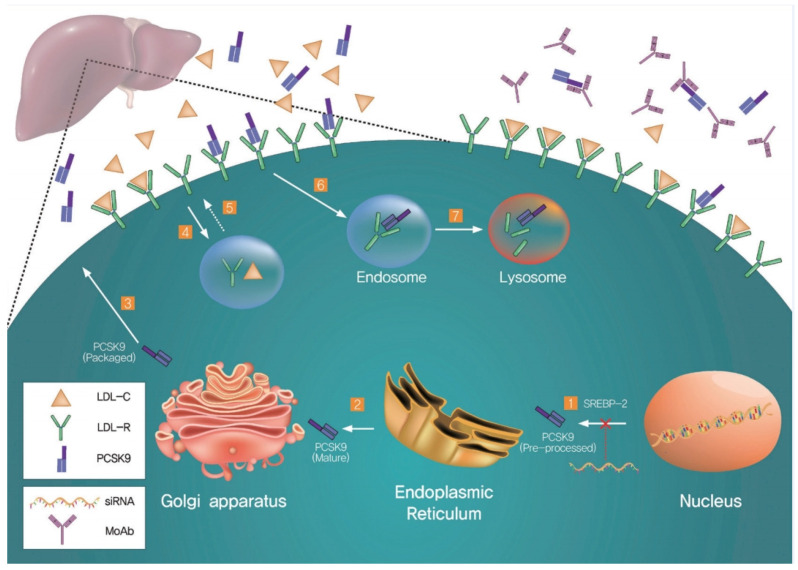
PCSK9-mediated regulation of LDLRs and representative therapy targets (Cho KH, Hong YJ, 2020, [55]). (1) SREBP-2 enhances transcription of both LDLRs and PCSK9. PCSK9 is processed in the ER (2) and stored in the Golgi apparatus (3) before secretion. LDLRs bind the LDL cholesterol (LDL-C) on the cell surface and this complex is internalized via endocytosis (4). LDL-C leaves LDLRs and LDLRs are recycled on the cell surface (5). Secreted PCSK9 binds to LDLRs and the complex is captured in the endosome (6) leading to LDLR lysosomal degradation (7). There are two major approaches to inhibit PCSK9: using siRNA and PCSK9-specific monoclonal antibodies (MoAbs). The siRNA inhibits PCSK9 translation (lower right), whereas MoAbs block the binding of PCSK9 to surface-expressed LDLRs (upper right). An especially attractive alternative would be the discovery of orally bioavailable low-molecular PCSK9 inhibitors which are economically more feasible than the mAbs.

**Figure 3 molecules-27-00434-f003:**
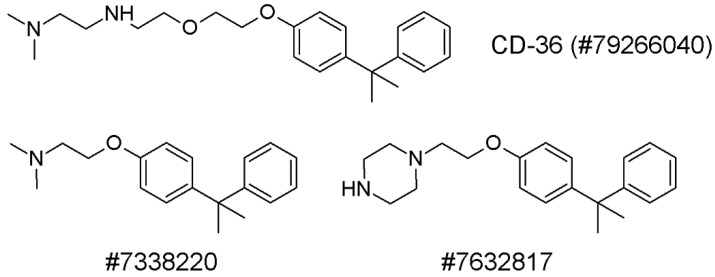
CB-36 and its analogues. The numbers are ChemBridge database identifiers.

**Figure 4 molecules-27-00434-f004:**
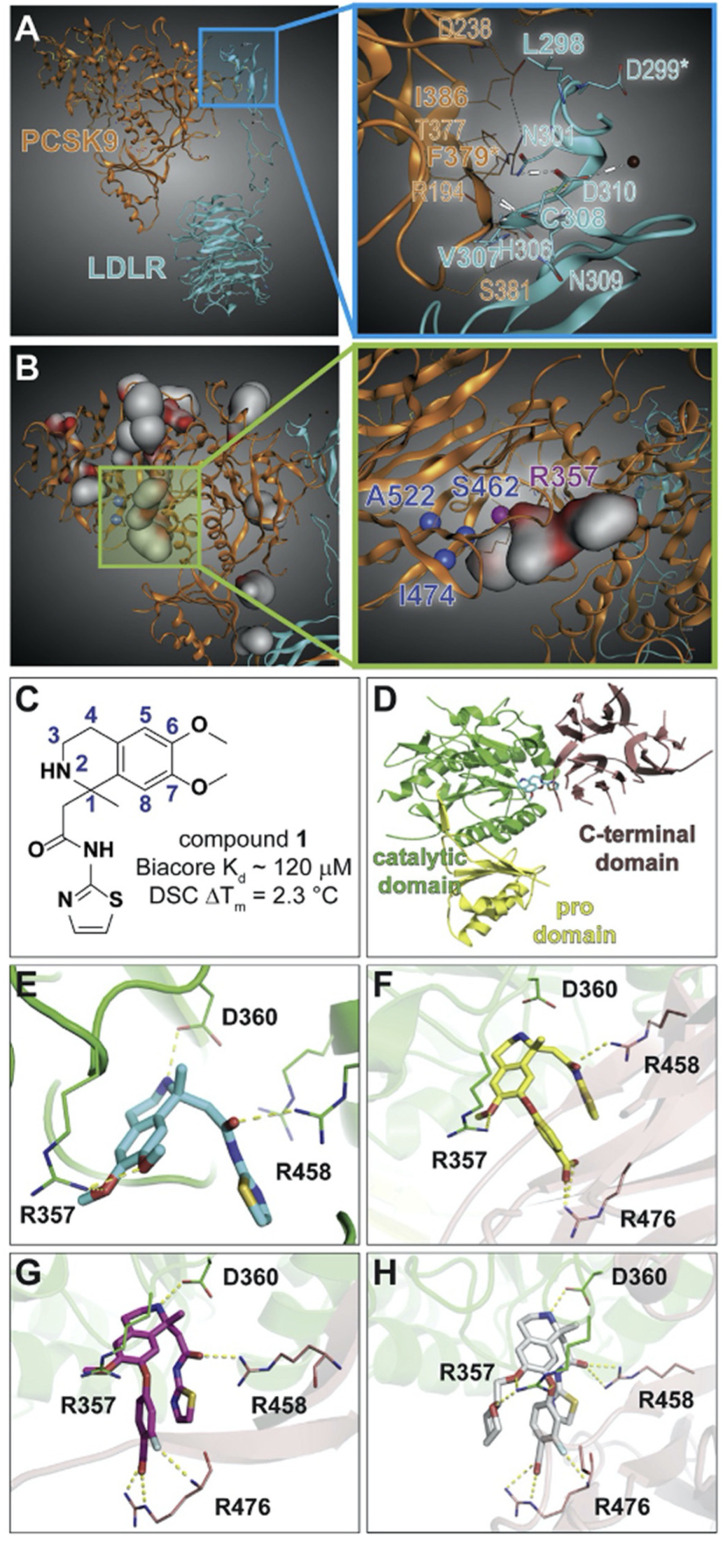
PCSK9/LDLR complex crystal structures with multiple small molecule ligands (Petrilli, W.L. et al., 2020 [59]). (**A**) Binding interface of PCSK9 (orange) and LDLR (sky-blue) (PDB ID 3P5B; Surdo et al., 2011). The box shows residues participating in the binding. The residues in bold are hydrophobic, and those marked by asterisks are the hotspots within 10 Å. (**B**) Binding site prediction using Molecular Operating Environment software package. In the box, a binding pocket is described which was detected next to the known human mutation. At the sites in blue, gain-of-function mutations were detected, whereas residues in purple could lead to function loss. The red spots on the surface denote the hydrophilic area. (**C**) Structural and binding data of compound **1**. (**D**) Crystal structure reveals that compound **1** binds to the interface between catalytic and C-terminal PCSK9 domains. (**E**) X-ray structure of compound **1**’s enantiomer bound to PCSK9. Its interactions with R357, D360, and R458 residues are visible (PDB: 6U3X). Cocrystallization of 1 with PCSK9 reveals that only its R-enantiomer (compound **2**) is mandatory, since the crystal structure of PCSK9-compound **2** was of higher resolution than that of the racemic mixture of **1**. (**F**) X-ray of compound **2** bound to PCSK9. The complex with carbonic acid 4 revealed a new interaction with R458 (PDB: 6U2N). (**G**) metafluorine of compound **5** is oriented to form a hydrogen bond with R476 (PDB: 6U2P). (**H**) Tetrahydropyran group at C6 of compound **8** is positioned to interact with R357 along with methoxy group oxygen of C7 (PDB: 6U38).

**Figure 5 molecules-27-00434-f005:**
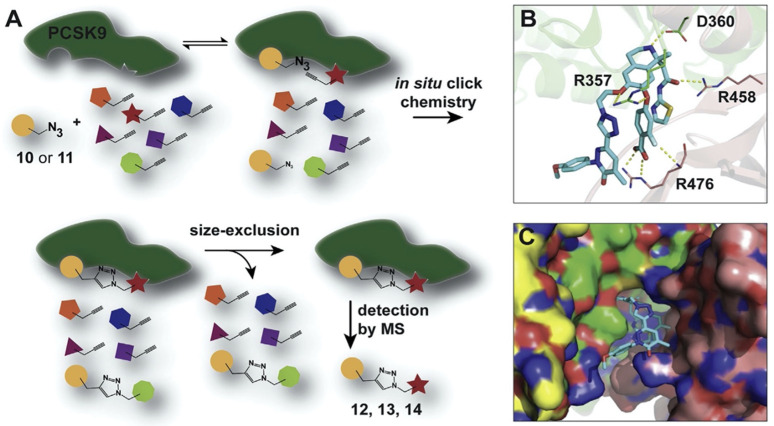
Illustration of PCSK9-mediated and -guided cycloaddition and X-ray complex with compound **14** (Petrilli, WL et al., 2020, [59]). (**A**) PCSK9 is incubated with modified azide, with several alkynes differing in molecular weight. If both azide and alkyne bind to the PCSK9 allosteric site in proximity to each other, 1,3-dipolar cycloaddition is promoted. Nonreacted compounds as well as nonbinding products were removed by size-exclusion chromatography, whereas MS was employed to determine the precise mass of the binder alkynes. (**B**) Specific interactions of compound **14** with PCSK9 at an allosteric binding site. (**C**) Spatial orientation of solvent-exposed substituent at C6 of compound **14** (PDB: 6U36). Interaction of pyranic oxygen with R357 can also be seen.

**Figure 6 molecules-27-00434-f006:**
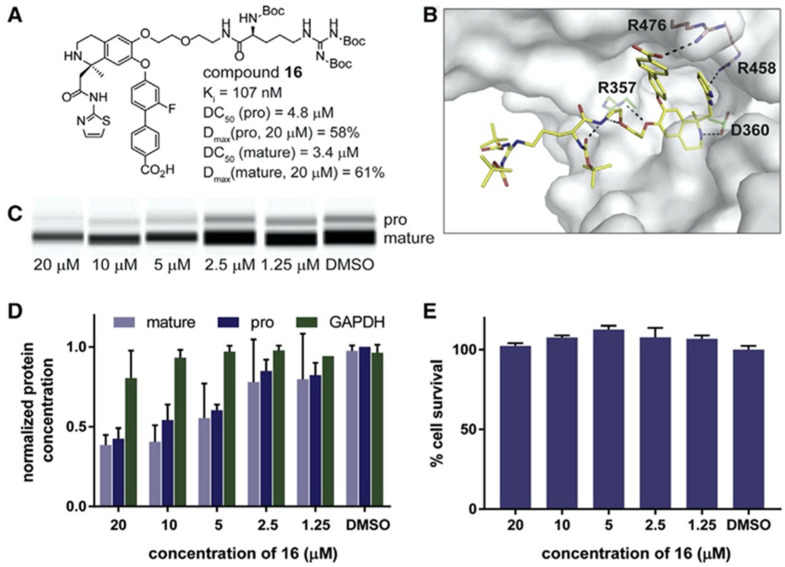
Targeted PCSK9 degradation in the HEK293 cells (Petrilli WL et al., 2020) [59]. (**A**) Compound **16** and its binding to PCSK9. (**B**) Crystal structure of compound **16** bound to PCSK9 (PDB: 6U26). (**C**) Western blot analysis of pro-PCSK9 and mature PCSK9 after 24 h of incubation with compound **16** in 1.25–20 mM of HEK293 cells, overexpressing PCSK9. (**D**) Analysis of overall PCSK9 and GAPDH levels after 48 h incubation with compound **16** in 1.25–20 mM of HEK293 cells, overexpressing PCSK9. The error bars denote SD. (**E**) Cell viability analysis (CellTiter-Glo) after 48 h incubation of HEK293 cells with 1.25–20 mM of compound **16**. Error bars denote SD.

**Table 1 molecules-27-00434-t001:** PCSK9 inhibitors and their mechanism of action.

Mechanism of Action	Examples
Blocking of PCSK9/LDLR complex formation	Peptides
	Adnectins
	Monoclonal antibodies
	(alirocumab, evolocumab, bocozicumab)
Suppression of PCSK9 expression	CRISPR/Cas9 technology
	Small molecules:
	berberine, oleanolic acid
	Antisense nucleotides
	siRNA
Interference with PCSK9 secretion	Sortilin
	Sec24a

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
