# Peer review of "PCSK9 as a Target for Development of a New Generation of Hypolipidemic Drugs"

_molecules, 2022, doi:10.3390/molecules27020434_

Round 1

Reviewer 1 Report

Thank you for submitting the manuscript “РСSК9 as a Target for Development of a New Generation of Hypolipidemic Drugs” to Molecules. Overall, the subject is interesting, but I have some questions. Overall, I felt confused as a reader of the article. Although the subject is interesting, in some parts of the work it gave the impression that it was not a review article, but a research one. It is necessary to evaluate what the authors actually did and if the objective is linked to a review of the relevant literature, the text must be adequate with appropriate citations. Also, there are several typos and formatting errors that need to be corrected. These small errors I ignored as the authors did not add lines to the manuscript it is difficult to point out where the error is in the revision.
- Introduction:
MI: first time the abbreviation appears. What does it mean?
T.B.: first time the abbreviation appears. What does it mean?
“The membrane-bound LDLR localized at clathrin-coated pits.” This sentence is incomplete.
- I didn't understand the need for items 5 and 6
- Figure 1 almost impossible to read due to low resolution
- “To achieve this, we made use of mass-spectrometry-based technology (AS / MS) to screen library over 200000 compounds large for PCSK9.” This was not a Review Article? It was what was proposed in the objective of the work.

Author Response

Point 1: MI: first time the abbreviation appears. What does it mean?

Response 1: MI means myocardial infarction. The correction has been submitted.

Point 2: T.B.: first time the abbreviation appears. What does it mean?

T.B. are Prof. Strom, MD, PhD initials

Point 3: “The membrane-bound LDLR localized at clathrin-coated pits.” This sentence is incomplete.

Response 3: The sentence has been completed.

Point 4. - I didn't understand the need for items 5 and 6

Response 4: The item 4 has been re-organized. The items 5 and 6 have been removed.

Point 5. - Figure 1 almost impossible to read due to low resolution

Response 5: The resolution of this figure has been improved.

Point 6. - “To achieve this, we made use of mass-spectrometry-based technology (AS / MS) to screen library over 200000 compounds large for PCSK9.” This was not a Review Article? It was what was proposed in the objective of the work.

Response 6: In fact, this is a review article. Unfortunately, we conceded several mistakes including this one which have been corrected consequently.

Reviewer 2 Report

The authors well described overall pictures regarding PCSK9 as a target molecule of future lipid-lowering therapy. The pivotal two randomized trials of PCSK9 inhibitor demonstrating significant reduction of cardiovascular events among patients with atherosclerotic cardiovascular disease; FOURIER trial; ODYSSEY Outcomes trial; should be briefly mentioned. I have no further comment.

Author Response

Point 1. The authors well described overall pictures regarding PCSK9 as a target molecule of future lipid-lowering therapy. The pivotal two randomized trials of PCSK9 inhibitor demonstrating significant reduction of cardiovascular events among patients with atherosclerotic cardiovascular disease; FOURIER trial; should be briefly mentioned. I have no further comment.

Response 2: The information of pivotal two randomized trials of PCSK9 inhibitor has been added.

Round 2

Reviewer 1 Report

Thank you for submitting your manuscript to Molecules. The authors made the requested corrections and my suggestion now is that the article be accepted for publication.